# Epidemiology and Burden of Diabetic Foot Ulcer and Peripheral Arterial Disease in Korea

**DOI:** 10.3390/jcm8050748

**Published:** 2019-05-25

**Authors:** Dong-il Chun, Sangyoung Kim, Jahyung Kim, Hyeon-Jong Yang, Jae Heon Kim, Jae-ho Cho, Young Yi, Woo Jong Kim, Sung Hun Won

**Affiliations:** 1Department of Orthopaedic Surgery, Soonchunhyang University Hospital Seoul, 59, Daesagwan-ro, Yongsan-gu, Seoul 04401, Korea; orthochun@gmail.com (D.-i.C.); hpsyndrome@naver.com (J.K.); 2SCH Biomedical Informatics Research Unit, Soonchunhyang University Seoul Hospital, Seoul 04401, Korea; kkimsy@naver.com; 3Department of Pediatrics, Soonchunhyang University Hospital Seoul, 59, Daesagwan-ro, Yongsan-gu, Seoul 04401, Korea; pedyang@schmc.ac.kr; 4Department of Urology, Soonchunhyang University Hospital Seoul, 59, Daesagwan-ro, Yongsan-gu, Seoul 04401, Korea; piacekjh@hanmail.net; 5Department of Orthopaedic Surgery, Chuncheon Sacred Heart Hospital, Hallym University, 77, Sakju-ro, Chuncheon-si 24253, Korea; hohotoy@nate.com; 6Department of Orthopaedic Surgery, Seoul Foot and Ankle Center, Inje University, 85, 2-ga, Jeo-dong, Jung-gu, Seoul 04551, Korea; 20vvin@naver.com; 7Department of Orthopaedic Surgery, Soonchunhyang University Hospital Cheonan, 31, Soonchunhyang 6-gil, Dongnam-gu, Cheonan 31151, Korea; kwj9383@hanmail.net

**Keywords:** diabetic foot ulcer, peripheral arterial disease, incidence, prevalence, cost, National Health Insurance Service data

## Abstract

Information about the epidemiology of diabetic foot ulcer (DFU) with peripheral arterial disease (PAD) is likely to be crucial for predicting future disease progression and establishing a health care budget. We investigated the incidence and prevalence of DFU and PAD in Korea. In addition, we examined costs of treatments for DFU and PAD. This study was conducted using data from Health Insurance Review and Assessment Service from 1 January 2011 to 31 December 2016. The incidence of DFU with PAD was 0.58% in 2012 and 0.49% in 2016. The prevalence of DFU with PAD was 1.7% in 2011 to 1.8% in 2016. The annual amputation rate of DFU with PAD was 0.95% in 2012 and 1.10% in 2016. Major amputation was decreased, while minor amputation was increased. The direct cost of each group was increased, especially the limb saving group. which was increased from 296 million dollars in 2011 to 441 million dollars in 2016. The overall incidence of DFU with PAD was about 0.5% of total population in Korea, from 2012 to 2016. Furthermore, costs for treatments of diabetic foot ulcer are increasing, especially those for the limb saving group.

## 1. Introduction

The prevalence of diabetes mellitus is expected to increase and the number of diabetic patients worldwide is on the rise. The global prevalence of diabetic foot varies from 3% in Oceania to 13% in North America, with a global average of 6.4% [1]. The annual incidence of diabetic foot ulcer (DFU) or necrosis in diabetic patients is known to be about 2% to 5% and the lifetime risk ranges from 15% to 20% [2,3,4]. Peripheral arterial disease (PAD), like cardiovascular disease, is a major arterial disease caused by atherosclerosis [5]. Diabetes is one of the high risk factors of PAD [5], and Olinic et al. [6] reported that the prevalence of PAD in Europe is increasing, parallel with increasing age and other risk factors for cardiovascular disease. PAD is associated with a 20-fold higher prevalence in patients with diabetes. It is known to be a risk factor for the highest severity of single factors in diabetic patients [7,8,9]. In addition, the probability of amputation within one year after the first ulcer or gangrene is 34.1% and the mortality rate has been reported to be 5.5% [8]. Information about the epidemiology of peripheral arterial disease associated with DFU is likely to be crucial for predicting future disease progression and establishing a health care budget.

About 20% to 33% of costs related to diabetes mellitus are used for treatments of diabetic foot [3,10]. The incidence of diabetes represented by chronic diseases is increasing. The cost of medical care for diabetic foot is increasing. Korea has recently entered an aging society. The increase in the number of diabetic patients has become an important issue in the decision of the health and welfare budget in Korea. In addition, the increase in complications due to diabetes is a burden, not only for patients, but also for the nation. Furthermore, such information is important for public health policy makers to advocate for implementation of prevention and treatment recommendations. However, there are no recent studies on the incidence, prevalence, or costs of treatments of DFU and PAD in Korea. 

Thus, the primary objective of this study was to investigate the incidence and prevalence of DFU and PAD in Korea. The secondary objective was to analyze the costs of treatments for DFU and PAD using National Health Insurance Service data provided by the Health Insurance Review and Assessment Service (HIRA). 

## 2. Materials and Methods

This study was approved by the Institutional Review Board of Soonchunhyang University Hospital Seoul (Institutional Review Board number: SCHUH 2018-01-007). The use of codes directly signifying DFU began on 1 January 2011, when the sixth edition of the Korean statistical classification of disease and related health problems-6 system (KCD-6) was applied. Until the year 2010, the disease code indicating the gangrene and ulcer was used separately from the diabetes code. If the disease code of the foot wound was not actively recorded, even if there was a DFU, DFU patients were inevitably missing. Therefore, we judged that it was not accurate to investigate data before 2010. Finally, data after 2011 were examined in this study. This study was conducted using data from HIRA from 1 January 2011 to 31 December 2016. 

The annual incidence and prevalence of diabetes foot ulcer and PAD among the total population of Korea (estimated population) reported by the National Statistical Office were calculated. We considered the wash-out period as one year to determine the annual incidence of DFU and PAD. Therefore, the annual incidence of newly diagnosed DFU and PAD patients was calculated from 2012. 

The amputation rates in diabetic foot ulcer and PAD patients were also calculated according to amputation level (minor vs. major (above ankle)). Diabetic foot ulcer and PAD codes and behavior codes (such as amputation, debridement, etc.) included in this study are summarized in Table 1. 

The direct cost for each amputation was calculated. We also analyzed the direct costs of DFU and PAD care in three groups. Group I was a limb-saving group. Group II was for those who had one amputation. Group III was for patients who had repeated amputation. The cost was based on the direct cost of patient contributions plus insurance claims. The direct cost was adjusted by taking into account the medical price index presented by the Korean Statistical Information Service (KOSIS, Daejeon, Korea). The data of this study were analyzed using SAS Enterprise Guide, ver. 6.1 M1 (SAS Institute Inc., Cary, NC, USA).

## 3. Results 

Regarding the overall annual incidence of DFU from 2012 to 2016, 0.43% of total populations were diagnosed with DFU in 2012 whereas 0.34% were diagnosed in 2016, showing a remarkable incidence plateau with a mild decrease over five years. The annual incidence of PAD was 0.19% in 2012 and 0.20% in 2016, showing an incidence plateau with a mild increase over five years. The annual incidence of DFU with PAD was 0.58% in 2012 and 0.49% in 2016 (Figure 1). The overall prevalence of DFU in the study period was 1.4% in 2011 and 1.3% in 2016. The prevalence of PAD was 0.4% in 2011 and 0.5% in 2016. The prevalence of DFU with PAD showed a mild increase from 1.7% in 2011 to 1.8% in 2016 (Figure 1). 

The annual amputation rate of DFU with PAD was increased from 0.95% in 2012 to 1.10% in 2016. Of these, the major amputation rate was decreased from 0.28% in 2012 to 0.27% in 2016, while the minor amputation rate was increased from 0.66% in 2012 to 0.82% in 2016 (Figure 2). 

The direct cost of amputation was increased from 17 million dollars in 2011 to 25 million dollars in 2016. Especially, the sum of direct costs of minor amputation increased from 11 million dollars in 2011 to 17 million dollars in 2016 (Figure 3). The average cost of amputation per person was also increased from 6100 dollars in 2011 to 7300 dollars in 2016 (Figure 4). The direct cost of each group was increased from 2011 to 2016. Direct costs for group 1 increased from 296 million dollars in 2011 to 441 million dollars in 2016. These costs for group 2 increased from 7.1 million dollars in 2011 to 9.3 million dollars in 2016, while those for group 3 increased from 10 million dollars in 2011 to 15 million dollars in 2016 (Figure 5).

## 4. Discussion

Overall incidence and prevalence of DFU with PAD in Korea from 2012 to 2016 were about 0.5% and 1.7% of the total population, respectively. The amputation rate was increased, especially the minor amputation rate, which increased from 0.66% in 2011 to 0.82% in 2016. Furthermore, direct costs for diabetes treatment were increased, especially the expense for the limb saving group.

We investigated the annual incidence and prevalence of DFU and PAD among the total population in Korea. The incidence and prevalence of DFU are also important for determining the number of diabetic patients. We requested HIRA to provide the total data for diabetic patients. However, the organization explained to us that these data were too large to release. Therefore, we could not obtain information for the total number of diabetic patients during the study period. Thus, we calculated the incidence and prevalence of DFU patients in the total population. However, the Korean diabetic association reported that the prevalence of diabetes increased from 12.4% in 2012 to 14.4% in 2016 through a diabetic fact sheet in 2018. Thus, we could investigate the prevalence of diabetic foot ulcer among the diabetic patients indirectly, showing 10% in 2012 and 9% in 2016. Recently, Zhang et al. [1] reported that the global prevalence of diabetic foot ulceration is 6.3% and the prevalence is 13.0% in North America and 5.5% in Asia. However, in their systematic review and meta-analysis study, the definitions for diabetic foot and diabetic foot ulceration were ambiguous. Furthermore, two epidemiologic studies using only Korean data focused on the epidemiology of diabetic peripheral neuropathy [11,12]. However, our study investigated not only DFU, but also PAD. Thus, we believe that our study has a more accurate prevalence of DFU in Korea. 

Concerning amputation, our results showed that annual amputation rate of DFU with PAD increased from 0.95% in 2012 to 1.10% in 2016. Of these, the major amputation rate decreased from 0.28% in 2012 to 0.27% in 2016, while the minor amputation rate was increased from 0.66% in 2012 to 0.82% in 2016. The overall amputation rate was increased. This might be due to the increased minor amputation rate, rather than the decrease of the major amputation rate. Goodney et al. reported that lower extremity amputation decreased by 45% from 1996 to 2011 (above the knee amputation decreased by 48% and below the knee amputation decreased by 39%) [13]. Although, in our study, we did not show a clear causal relationship about the reason for this situation, two reasons might be important. The first one is the increased awareness of the risk of diabetic foot in diabetic patients. Previous studies have reported that education about foot care to diabetic patients is important because it is associated with a significant reduction in lower extremity amputation. In addition, monthly foot checks are associated with the reduction of major lower limb amputations in diabetic incident hemodialysis patients [14,15]. Future studies are needed to examine awareness of risk of DFU in diabetic patients. The second reason is the improvement of vascular conditions due to increase of revascularization. This is also important for the reduction of major amputation. Peripheral vascular disease is known to be the most significant risk factor for diabetic foot amputation [8]. A previous study also reported that it is evident that the increasing use of vascular and preventive care, especially among patients with diabetes, is temporally associated with lower rates of major amputation [16]. Future studies focusing on understanding this relationship are needed. 

The increase in the cost of medical care, due to the increased number of diabetic patients, has been reported all over the world. It is a useful indicator for planning and enforcing health care policies and budgets [17,18,19]. To the best of our knowledge, our research is the first to investigate the cost of medical care for diabetic foot ulcer in Korea. Our study showed that, although the cost of amputation was increased a lot, the expense for the limb saving group increased exponentially. Such a cost increase might be due to the development of medications and dressing materials. This could increase the burden on the patient. Understanding the cost of DFU should support future decisions on investment in diabetic foot care. 

Some limitations of the study need to be addressed. First, DFU and PAD codes are diverse, unclear, and sometimes missing. There was no defined disease code for DFU until 2010. Thus, data on DFU from 2007 to 2010 could not be used. This is considered the limit for using National Health Insurance Service data provided by HIRA. It is necessary to agree on codes of diabetic foot and PAD more clearly and uniformly in the future. Second, diabetic neuropathy was not included in this study, because the disease code and operational definition for diabetic neuropathy were not established in the Big Data. Furthermore, extracted data using the provided diabetic neuropathy code revealed that too many patients were included, so the extracted data could not be trusted. Therefore, in this study, diabetic neuropathy was excluded in order to improve the quality of the study, but it is thought that studies to include neuropathy in the diabetic foot should be done through the operational definition defined later. Third, when calculating the incidence of chronic diseases, such as diabetes and PAD, a wash-out period of at least 2 years should be used. However, we used wash-out period of one year to calculate incidence of DFU and PAD. This was an inevitable choice due to too much missing data. If data for a longer period of time can be used, the wash-out period of 2 years can be used. Such study is needed in the future. 

## 5. Conclusions

In conclusion, over 5 years, we found that the overall incidence and prevalence of DFU with PAD in Korea were about 0.5% and 1.7% of the total population, respectively. The amputation rate was increased, especially the minor amputation rate. Furthermore, the direct costs for DFU treatment were increased, especially the expense for the limb saving group. Our results suggest that we should pay attention to effective implementation of the budget when we make future health policies for diabetic foot. Further studies warrant the importance of productive and cost-effective methods for saving limbs in the healthcare system. 

## Figures and Tables

**Figure 1 jcm-08-00748-f001:**
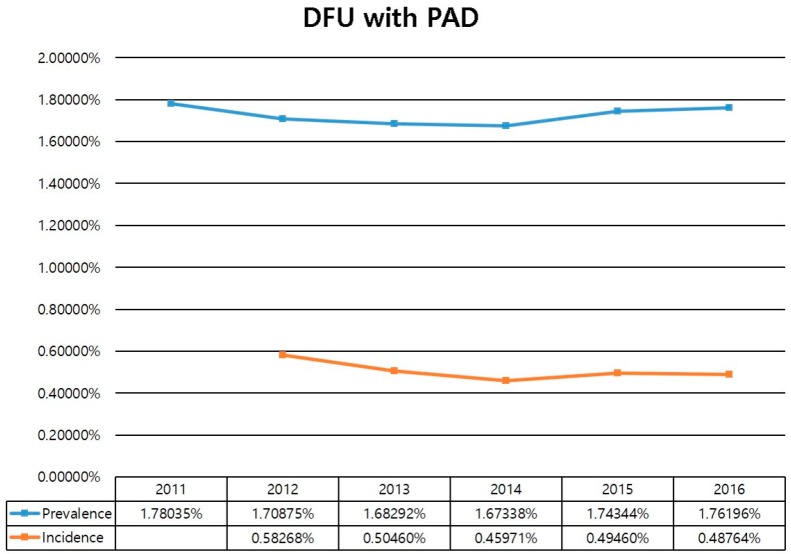
Annual the incidence and prevalence of diabetic foot ulcer with PAD.

**Figure 2 jcm-08-00748-f002:**
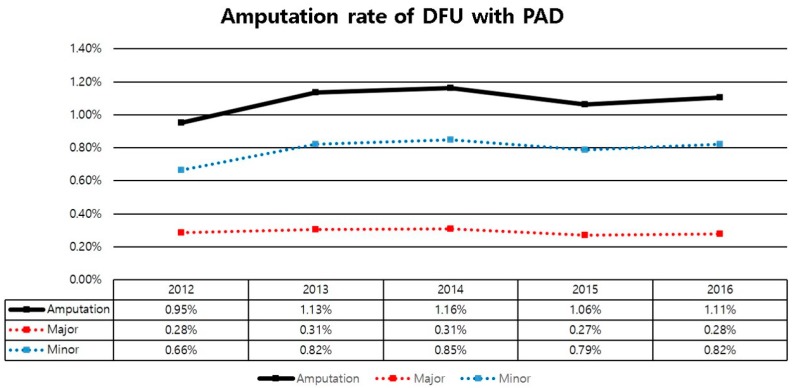
Annual amputation rate of diabetic foot ulcer with PAD.

**Figure 3 jcm-08-00748-f003:**
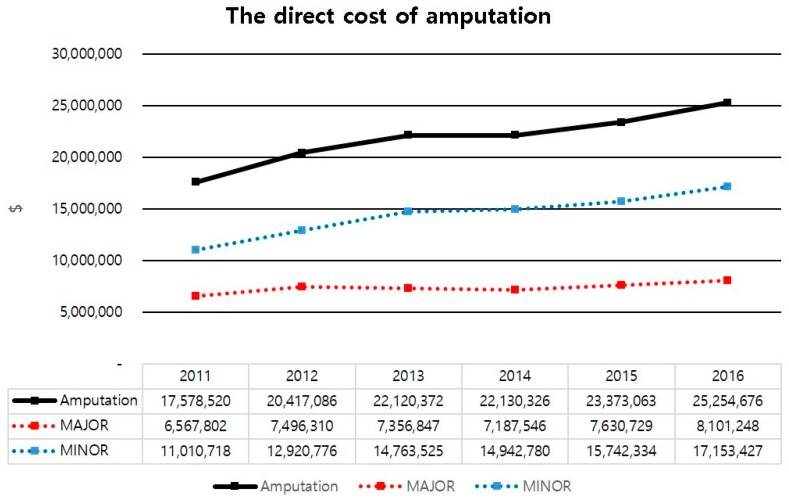
The direct cost of amputation of diabetic foot ulcer with PAD.

**Figure 4 jcm-08-00748-f004:**
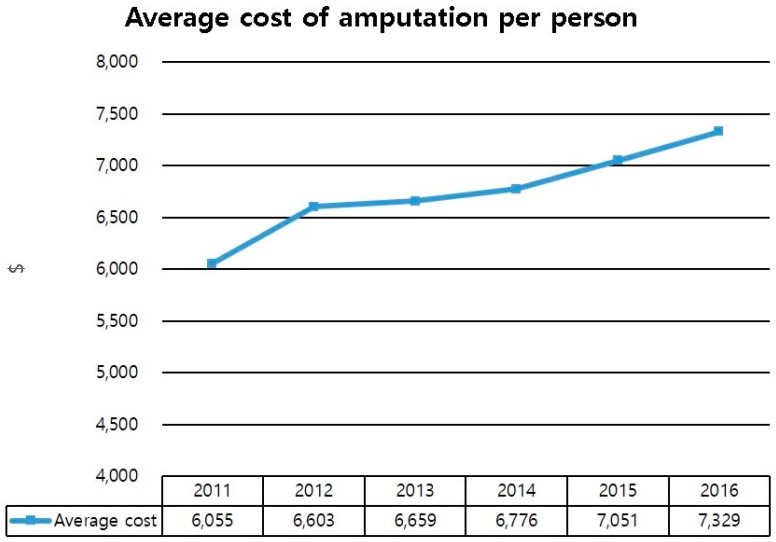
The average cost of amputation, per person, of diabetic foot ulcer with PAD.

**Figure 5 jcm-08-00748-f005:**
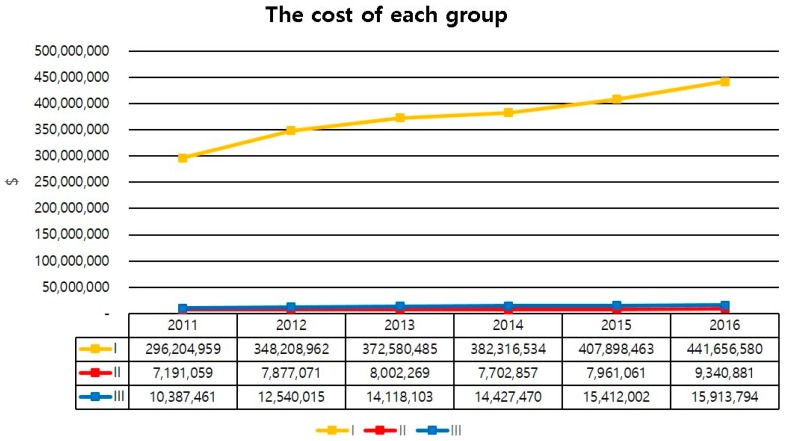
The direct cost of each group for diabetic foot ulcer with PAD.

**Table 1 jcm-08-00748-t001:** Diabetic foot ulcer (DFU) and peripheral arterial disease (PAD) codes and behavior codes included in this study.

**Diabetic Foot**
E105: E1050, E1051, E1058
E107: E1070, E1071, E1072, E1078
E115: E1150, E1151, E1158
E117: E1170, E1171, E1172, E1178
E125: E1250, E1251, E1258
E127: E1270, E1271, E1272, E1278
E135: E1350, E1351, E1358
E137: E1370, E1371, E1372, E1378
E145: E1450, E1451, E1458
E147: E1470, E1471, E1472, E1478
**Peripheral Arterial Disease**
I7022, I7023, I7024, I7025, I7029
**Behavior**
SC021, SC022, SC023, SC024, SC025, SC026, SC027
M0111, M0115, M0121, M0122, M0123, M0125, M0135, M0137
N0571, N0572, N0573, N0574, N0575, N0579

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
