# Peer review of "Epidemiology and Burden of Diabetic Foot Ulcer and Peripheral Arterial Disease in Korea"

_jcm, 2019, doi:10.3390/jcm8050748_

Round 1
Reviewer 1 Report
The topic of this manuscript falls within the scope of Journal of Clinical Medicine. The topic of the manuscript is very important. The number of patients with PAD a
The topic of this manuscript falls within the scope of Journal of Clinical Medicine. The topic of the manuscript is very important. The number of patients with PAD and diabetic foot syndrome is still increasing. What is more, the number of lower limb amputations is also increasing. The text is clear and easy to read. This study was 30 conducted using data from Health Insurance Review and Assessment Service from January 1, 2011 31 to December 31, 2016.The Author showed that overall incidence and prevalence of diabetic foot ulcer with PAD 194 over 5 years in Korea were about 0.5% and 1.7% of the total population, respectively. The amputation 195 rate was increased, especially the minor amputation rate which was increased. Furthermore, direct 196 costs for diabetic foot ulcer treatment were increased, especially expense for the limb saving group. The conclusions are consistent with presented evidence and arguments.
There are only few comments in the reviewer opinion which should be taken under consideration by the authors:
1. In the introduction or discussion please include the newest paper, which presents the epidemiological data of PAD and diabetic foot syndrome in Europe “Olinic D.M et al. Epidemiology of peripherial disease in Europe- VAS Educational Paper. International angiology, 2018,37, 4, 327-334”; Gębala-Prajsnar K., et al. Selected physical medicine interventions in the treatment of diabetic foot syndrome. Acta angiologica, 2015, 21,4,140-145 doi 10.5603/AA.2015.0024; Behrend CA et al. International Variations in Amputation Practice: A VASCUNET Report. Eur J Vasc Endovasc Surg. 2018 Sep;56(3):391-399.
2. In the discussion It would be worth pointing out that PAD is an independent risk factor for increased cardiovascular mortality and that most relevant comorbidities in patients with PAD are CAD and diabetes mellitus (DM). Patients with diagnosed DM are characterized by higher frequency of PAD occurrence Wojtasik-Bakalarz J, et al. Impact of coronary artery disease and diabetes mellitus on the long-term follow-up in patients after retrograde recanalization of femoropopliteal arterial region. Journal of Diabetes Research, vol.2019, Article ID 6036359, 6 pages, 2019
Author Response
There are only few comments in the reviewer opinion which should be taken under consideration by the authors:
1. In the introduction or discussion please include the newest paper, which presents the epidemiological data of PAD and diabetic foot syndrome in Europe “Olinic D.M et al. Epidemiology of peripherial disease in Europe- VAS Educational Paper. International angiology, 2018,37, 4, 327-334”; Gębala-Prajsnar K.et al. Selected physical medicine interventions in the treatment of diabetic foot syndrome. Acta angiologica, 2015, 21,4,140-145 doi 10.5603/AA.2015.0024; Behrend CA et al. International Variations in Amputation Practice: A VASCUNET Report. Eur J Vasc Endovasc Surg. 2018 Sep;56(3):391-399.
Response: We appreciate your comment. We add upper recommended original articles on introduction part as followings; “and Olinic D.M et al [6] reported that the prevalence of PAD in Europe is increasing, parallel with increasing age and other risk factors for cardiovascular disease.”
2. In the discussion It would be worth pointing out that PAD is an independent risk factor for increased cardiovascular mortality and that most relevant comorbidities in patients with PAD are CAD and diabetes mellitus (DM). Patients with diagnosed DM are characterized by higher frequency of PAD occurrence Wojtasik-Bakalarz J, et al. Impact of coronary artery disease and diabetes mellitus on the long-term follow-up in patients after retrograde recanalization of femoropopliteal arterial region. Journal of Diabetes Research, vol.2019, Article ID 6036359, 6 pages, 2019
Response: We appreciate your comment. We totally agree with your opinion. However, we thought that it would be better upper recommended topic was be in the ‘introduction part’. Therefore, we deal with upper topic on introduction part, and add reference that you recommended as followings; It is known to be a risk factor for the highest severity of single factors in diabetic patients[7-9].
Reviewer 2 Report
The main objective of this study was to examine the incidence and cost of Diabetic Foot Ulcer (DFU) and peripheral arterial disease (PAD) to the healthcare system in Korea using the data from HIRA between 2011 and 2016. The incidence and cost of DFU is varying over the years due to reporting system in countries around the world. It is important to design an epidemiology study in DFU incidence and cost but without any statistical analysis, it is sub-par. In lines 93-94, authors stated that the study were analyzed using a software but I did not see any statistical analysis. Also, I would encourage authors to incorporate following findings throughout the manuscript for further consideration.
Title: Please remove “Running Title” from the title.
Abstract: Please abbreviate Diabetic Foot Ulcer as (DFU) for the first time usage then implement this change throughout the manuscript.
Please add a conclusion statement in addition to findings. This statement may include “further studies warrant the importance of productive and cost-effective methods for saving limbs to the healthcare system”.
All figures (fig 1 to fig 5) please mark the data points per years then connect the points.
Figure5. Please define A, B and C in figure description. Lines 114-115 indicate as I, II and III.
Author Response
The main objective of this study was to examine the incidence and cost of Diabetic Foot Ulcer (DFU) and peripheral arterial disease (PAD) to the healthcare system in Korea using the data from HIRA between 2011 and 2016. The incidence and cost of DFU is varying over the years due to reporting system in countries around the world. It is important to design an epidemiology study in DFU incidence and cost but without any statistical analysis, it is sub-par. In lines 93-94, authors stated that the study were analyzed using a software but I did not see any statistical analysis. Also, I would encourage authors to incorporate following findings throughout the manuscript for further consideration.
Response: We appreciate your comment. We totally agree with your opinion. Our study used the HIRA data to investigate the incidence, prevalence and costs of diabetic foot and peripheral arterial disease of Big Data. The HIRA offers rawdata with millions of rows and a capacity of 300GB. This rawdata alone cannot identify the prevalence and incidence, so it is necessary to handle the rawdata using statistical package such as SAS.
Title: Please remove “Running Title” from the title.
Response: We appreciate your comment. We remove the “Running title”.
Abstract: Please abbreviate Diabetic Foot Ulcer as (DFU) for the first time usage then implement this change throughout the manuscript.
Response: We appreciate your comment. We change followings diabetic foot ulcer to DFU.
Please add a conclusion statement in addition to findings. This statement may include “further studies warrant the importance of productive and cost-effective methods for saving limbs to the healthcare system”.
Response: We appreciate your comment. We totally agree with your opinion. We add recommended comments followings on conclusion; Our results suggest that we should pay attention to effective implementation of the budget when we make future health policy of diabetic foot, further studies warrant the importance of productive and cost-effective methods for saving limbs to the healthcare system.
All figures (fig 1 to fig 5) please mark the data points per years then connect the points.
Response: We appreciate your comment. We changed figures as you recommend.
Figure5. Please define A, B and C in figure description. Lines 114-115 indicate as I, II and III.
Response: We appreciate your comment. We change the indicator A to I, B to II and C to III.
Round 2
Reviewer 2 Report
I would to thank authors for appropriate edits.